# Biofilm Formation in Clinical Isolates of *Fusarium*

**DOI:** 10.3390/jof10110766

**Published:** 2024-11-04

**Authors:** Ray Zhang, Nathan Wiederhold, Richard Calderone, Dongmei Li

**Affiliations:** 1Department of Microbiology & Immunology, Georgetown University Medical Center, Washington, DC 20057, USA; 2025rzhang@tjhsst.edu (R.Z.); calderor@georgetown.edu (R.C.); 2Thomas Jefferson High School for Science and Technology, Alexandria, VA 22312, USA; 3Fungus Testing Laboratory, Department of Pathology and Laboratory Medicine, University of Texas Health Science Center at San Antonio, San Antonio, TX 78229, USA; wiederholdn@uthscsa.edu

**Keywords:** *Fusarium*, species complexes, fusariosis, biofilm biology, antifungal resistance

## Abstract

Many microbial pathogens form biofilms, assemblages of polymeric compounds that play a crucial role in establishing infections. The biofilms of *Fusarium* species also contribute to high antifungal resistance. Using our collection of 29 clinical *Fusarium* isolates, we focused on characterizing differences in thermotolerance, anaerobic growth, and biofilm formation across four *Fusarium* species complexes commonly found in clinical settings. We investigated the role of carbon sources, temperature, and fungal morphology on biofilm development. Using fluorescence microscopy, we followed the stages of biofilm formation. Biofilms were screened for sensitivity/resistance to the antifungals voriconazole (VOR), amphotericin B (AmB), and 5-fluorocytosine (5-FC). Our findings revealed generally poor thermotolerance and growth under anaerobic conditions across all *Fusarium* species. VOR was more effective than AmB in controlling biofilm formation, but the combination of VOR, AmB, and 5-FC significantly reduced biofilm formation across all species. Additionally, *Fusarium* biofilm formation varied under non-glucose carbon sources, highlighting the species’ adaptability to different nutrient environments. Notably, early stage biofilms were primarily composed of lipids, while polysaccharides became dominant in late-stage biofilms, suggesting a dynamic shift in biofilm composition over time.

## 1. Introduction

The filamentous fungal genus *Fusarium* is a member of the “big five” mold killers of humans and is also referred to as “Trans-Kingdom Pathogens” since it causes diseases in plants [1]. Recently, *Fusarium* spp. have been recognized as a high-priority group on the 2022 WHO Fungal Priority Pathogens List [2]. In the agricultural sector, essential crops, including wheat, bananas, and rice, ensure food security for millions but are threatened by this fungal genus through either mycotoxins or infections [3,4]. For instance, mycotoxin-producing *Fusarium* spp. cause blight and wilt in crops that result in a loss of 220 billion USD in agricultural revenue annually [5] and jeopardize human health when consumed [6].

Over 70% of all clinical *Fusarium* diseases are thought to occur through contact with air-borne or water-borne spores in the environment. Air-borne, inhaled spores can cause respiratory and/or disseminated diseases in otherwise healthy or, more usually, immunocompromised patients [7]. For *Fusarium* infections occurring in immunocompromised patients, mortality rates can be greater than 75% [8]. Globally, *Fusarium* spp. have emerged as the second most common mold fungal pathogen [9,10]. They remain 1 of the top 11 most dangerous fungal pathogens [2,11,12]. The disease spectrum of *Fusarium* spp. in clinical settings also includes broadly superficial and/or locally invasive infections, such as keratitis, onychomycosis, or sinusitis other than invasive disease [13].

Currently, the *Fusarium* genus consists of over 200 species that are taxonomically grouped within 22 species complexes, of which 74 of these species cause human infections [14]. Among the most clinically relevant species complexes are the *Fusarium solani* species complex (FSSC) and the *Fusarium oxysporum* species complex (FOSC), which account for 60% and 20% of all human *Fusarium* infections, respectively [13]. The third most commonly encountered species, *Fusarium fujikuroi*, is primarily a plant pathogen that causes seedling disease in rice, sugarcane, and asparagus. This species poses a serious threat to food and feed safety through carcinogenic Fumonisin production [15].

Like *F. fujikuroi*, the pathogens *F. solani* and *F. oxysporum* also produce mycotoxins—fumonisin and gibberellin—during plant infection, as well as other inflammatory proteins such as RPB2, TEF1α, and the AFLP marker EATMCAY107 [15]. Specifically, *F. fujikuroi* synthesizes over 1194 secretory proteins, of which 38% contribute to fungal virulence, including mitogen-activated protein kinases (MAPK), which results in cell wall perforation and the invasion of infected plants [16,17]. Another *Fusarium* species complex of interest is *Fusarium incarnatum-equiseti*, which is responsible for vomiting and diarrhea through toxic secondary metabolites such as nivalenol (NIV), deoxynivalenol (DON), and diacetoxyscirpenol (DAS) [18].

Biofilm production is another key virulence factor in *Fusarium*. Biofilms are a collection of both living and dead microbial cells embedded on a surface and enclosed by an extracellular polymeric matrix composed primarily of water, polysaccharides, proteins, and DNA [19]. The 3D structure of biofilms offers advantages to microorganisms during disease development, such as a strong surface adhesion to host cells, social interactions, initiation of cell–cell communication, and spatial organization [20]. Of particular interest, the biofilm extracellular matrix (ECM) impairs the chemical and physical penetration of anti-fungal drugs. Indeed, compared to free-living planktonic cells, bacterial biofilms display a 10-to-10,000-fold reduction in susceptibility toward antimicrobial drugs [21].

Prior to the work detailed here, many publications have only attempted to describe the adherence of *Fusarium* biofilms to various surfaces, with little detail on biofilm structure, development, or extracellular polymeric substance (EPS) components of *Fusarium* biofilms. A study characterizing the biofilm formation and structure of *Fusarium graminearum* found that biofilms are surrounded by a polymeric matrix that consists primarily of polysaccharides and extracellular nucleic acids but lacks lipids [22]. However, since antifungal susceptibility and extracellular matrix development vary across species complexes, they pose a major challenge in applying the results of one species to that of another [23].

To the best of our knowledge, no research exists comparing biofilm structures and polymeric compositions across different species complexes of *Fusarium*, nor on the combined activity of polymer-targeted antifungal agents to interfere with *Fusarium* biofilms. Thus, in this study, we investigated the specific macromolecular composition of *F. oxysporum*, *F. solani*, *F. fujikuroi*, and *F. incarnatum-equiseti* biofilms during their development in vitro, in particular focusing on the composition of biofilms at different stages of development and under an antifungal drug cocktail containing voriconazole (VOR), amphotericin-B (AmB), and 5-fluorocytosine (5-FC). The four species studied are the most economically and clinically important pathogens within the *Fusarium* genus. We further compare the biofilm formation of five *F. solani* strains with varying antifungal resistance profiles, a dominant species in our collection, to evaluate the effect of antifungal agents on preventing biofilm formation between resistant and susceptible *F. solani* isolates.

## 2. Materials and Methods

### 2.1. Fungal Strains and Culture Conditions

A total of 29 *Fusarium* clinical isolates were used in this study, provided by two sources. Seventeen isolates of *Fusarium solani* (A1–A17) were provided by CDC’s Mycotic Disease Branch laboratory, and twelve isolates, representing several *Fusarium* species, were provided by Dr. Nathan P. Wiederhold from the University of Texas Health Science Center at San Antonio. The latter collection (B1–B12) includes six isolates from the *F. fujikuroi* species complex (FFSC), four isolates of the *F. solani* species complex (FSSC), and one strain each from the *F. oxysporum* species complex (FOSC) and the *F. incarnatum-equiseti* species complex (FIESC). To better understand the effect of drug resistance in biofilm development, we included five *F. solani* strains with variable antifungal susceptibilities to common drugs for the biofilm study.

Long-term spore stocks were maintained as colonized blocks in potato dextrose broth (PDB; MP Biomedical, Irvine, CA, USA) with 15% glycerol at −80 °C. To establish a spore bank for further studies, *Fusarium* strains were cultivated via subculture on potato dextrose agar (PDA; Sigma-Aldrich, Darmstadt, Germany) at 30 °C for 6 days or otherwise indicated. A small portion of the growth culture on PDA (0.5 cm^2^) was transferred to the center of a PDA plate for growth curve measurements.

For this study, a spore suspension was prepared by scraping the surface of the mold growth on PDA with a sterilized blade in 5 mL of 2% Tween 20 (polysorbate 20 with Tris Buffered Saline; pH 7.4). Conidial spore suspensions were centrifuged at 3000 rpm for 10 min (Beckman GS-6R Centrifuge; Marshall Scientific, Hampton, NH, USA) and washed in 70% ethanol. Spores were re-suspended in RPMI-1640 (Sigma-Aldrich) containing 2 mM of L-glutamine, 10 mM of HEPES, and 1 mM of sodium pyruvate, then supplemented with 2% dextrose (D-glucose anhydrous; Sigma-Aldrich) and MOPS (3-(N-morpholino propanesulfonic acid; Fisher Scientific, Waltham, MA, USA) buffered to pH 7.0. The optical density of the spore suspension at 530 nm was measured by using a UV-Visible spectrophotometer (Dynex Technologies, Chantilly, VA, USA). In all experiments, the final spore stock concentration was 1 × 10^6^ conidia/mL.

### 2.2. Optimal Growth Temperature

Tolerance to high temperatures is a key pathogenic trait of many human pathogens. Growth at 37 °C is essential for the survival and disease progression of pathogenic fungi in humans, particularly for invasive fungi like *Candida albicans* and “Trans-Kingdom” mold pathogens. A recent study demonstrated that surface-exposed and charged residues contribute to fungal thermostability [24]. To assess the growth of different *Fusarium* species at 30 °C and 42 °C, the colony diameter (cm) of the 29 clinical isolates on PDA plates was measured every other day over 9 days. This method, commonly used in mold fungi, avoids inaccuracies from mycelial clumping in broth cultures when measured by OD read. The inoculated Petri dishes were incubated at the two temperatures in parallel using identical incubators (Fisher Scientific, Waltham, MA, USA). Growth curves were generated from the colony diameter measurements for analysis, with each strain tested in replicate at both temperatures.

### 2.3. Mature Colony Morphology and Anaerobic Stress Resistance Analysis

Fungal isolates were cultured from long-term spore stocks, as previously described, to develop colonies for anaerobic growth analysis. Small disks (0.5 cm^2^) of 6-day-old fungal strains were transferred to the center of PDA Petri dishes. The isolates were incubated at 30 °C under anaerobic conditions using an AnaeroPack Rectangular Jar 2.5 L System (Mitsubishi Gas Chemical Co., Tokyo, Japan). Controls were grown under aerobic conditions in Oxygen Demand Incubator at the same temperature. Colony diameters (cm) were measured at 8 days post-inoculation in triplicate experiments. The colonies formed in both temperatures were photographed to generate both front and back views.

Microscopic images were taken using an Olympus BH-2 microscope, Olympus America, Inc. Melville, NY, USA. Each isolate was morphologically classified by its quantity of characteristic macroconidia (multiple cells), microconidia (single cell), and branched hyphae.

### 2.4. Biofilm Formation and Varying Sugar Source

To characterize the rate of biofilm formation, a biofilm quantification assay using a modified crystal violet (CV) assay was adapted from the protocols described by Kischkel et al. [25], O’Toole [26], and Shay et al. [22]. First, *Fusarium* biofilms were developed in 96-well microtiter plates by pipetting 200 µL of the stock inoculate solution (1 × 10^6^ conidia/mL) into each well. The plates were incubated under static conditions at 37 °C for 24, 48, and 72 h. After incubation, culture wells were aspirated and washed twice with sterile distilled water to remove non-adherent and planktonic cells. Samples were allowed to air dry in a laminar hood. The EPS structure of the biofilm matrix was stained with 200 µL per well of 1% (*v*/*v*) CV (Sigma Chemical CO, St. Louis, MO, USA) for 10 min. Microtiter plates were then gently rinsed twice with sterile distilled water and completely dried. For biofilm quantification, the bound CV was de-stained with 200 µL per well of 30% (*v*/*v*) acetic acid (Carolina Biological Supply, Burlington, NC, USA) at room temperature (22–25 °C) for 10 min. Then, 100 µL of the supernatant was transferred to a new flat-bottomed microtiter plate. Absorbance at 595 nm was measured through a TRIAD Series Multimode Detector plate reader (Dynex Technologies), and optical density (OD_595_) was recorded.

The OD_595_ reads from experimental wells were evaluated by subtracting the background optical density of CV-stained control wells containing RPMI-1640 medium without *Fusarium* conidia. A strain was considered a proficient biofilm producer if its OD value exceeded 0.25, calculated as previously described for *Staphylococcus aureus* and *Fusarium* spp. [27,28]. Biofilm quantification assays were performed 3 times with 8 replicative wells for each strain and controls.

Incubation media with varying sugar sources were prepared in RPMI-1640 supplemented with varying sugar sources: (a) 2% glucose (control), (b) 0.2% glucose, (c) 2% D(+) xylose (Sigma-Aldrich), (d) 2% D(+) trehalose (α-D-glucopyranosyl α-D-glucopyranoside dihydrate; Sigma-Aldrich), (e) and 4 mL of 50% glycerol (Fisher Scientific). Subsequently, 100 µL of each medium was dispensed into wells of a microtiter plate with 100 µL of stock spore suspensions. Plates were incubated for 24 h at 30 °C to assess biofilm formation. Biofilm growth was quantified using the previously described biofilm CV assay. Four replicates for each isolate in each sugar medium were conducted.

### 2.5. Measurements of Biofilm Composition

*Fusarium* spp. biofilms were formed in 96-well Nunclon Delta Surface microtiter plates (Thermo Fisher Scientific, Waltham, MA, USA), as previously described for 6, 12, and 24 h at 37 °C [22]. Prior to staining, wells were rinsed with 1× PBS (0.02 M of potassium phosphate, 0.15 M of sodium chloride; pH 7.2; Rockland Immunochemicals, Pottstown, PA, USA), and supernatants were removed. Macromolecule-specific fluorochrome stains were utilized to identify biofilm matrix composition. Four fluorochromes were prepared by dissolving in dimethyl sulfoxide (DMSO; Sigma-Aldrich, St. Louis, MO, USA): 10 mg/mL of Calcofluor White (CFW; Fluorescent Brightener 28; Sigma-Aldrich, St. Louis, MO, USA), which stains chitin, cellulose, and polysaccharides; 10 mg/mL of Hoechst 33342 (Cell Signaling Technology, Danvers, MA, USA), which stains extracellular DNA in living and fixed cells; and 2 mg/mL of Rhodamine 123 (Rhodamine B; Enzo Biochem Inc., Farmingdale, NY, USA), which stains proteins; and 500 μg/mL Nile Red (NR; Sigma-Aldrich), which stains lipids. Biofilm samples were stained with 5 µL of each fluorochrome in the dark for 10 min at room temperature. Stained wells were washed once with 1 × PBS, and fluorescent intensities were measured using a microplate fluorescence reader (TRIAD Series Multimode Detector plate reader, Dynex Technologies, Chantilly, VA, USA) with the following fluorescence filters (excitation—emission): CFW and Hoechst 33342 (350–460 nm), rhodamine 123 (500–535 nm), and Nile Red (560–630 nm). Experiments were repeated three times on three independent days.

### 2.6. Fluorescence Visualization of Biofilm Composition

*Fusarium* spp. biofilms were formed on sterile polystyrene coverslips in 24-well microtiter plates, as previously described. Following incubation at 37 °C for 24 h, coverslips were removed from wells and rinsed with sterile distilled water to remove planktonic cells. Mature biofilms were stained with 5 µL of each aforementioned fluorochrome in the dark for 10 min at RT. Then, samples were rinsed with sterile distilled water and fixed with paraformaldehyde (0.4 M PHEM buffer, 16% paraformaldehyde stock, 1 × PBS; pH 7.4) for 2 h in the dark. Images were taken under a multichannel fluorescence imaging microscope (EVOS FL Auto, Life Technologies, Carlsbad, CA, USA) with the previously mentioned excitation and emission filters. All fluorescent images were quantitively analyzed using ImageJ 1.53e (NIH, Bethesda, MD, USA).

### 2.7. Antifungal Agent Preparation

Three antifungal agents were evaluated for their biofilm-inhibition properties: voriconazole (VOR; Pfizer Inc., New York, NY, USA), amphotericin B (AmB; A-2411; Sigma-Aldrich), and 5-fluorocytosine (5-FC; F-7129; Sigma-Aldrich). Stock antifungal solutions were prepared at a concentration of 5000 µg/mL in DMSO solution. These solutions were further diluted to 128 µg/mL in RPMI-1640 medium for use in subsequent antifungal susceptibility assays.

### 2.8. Antifungal Susceptibility Assays

The *Fusarium* antifungal inhibition assay was performed according to the broth dilution antifungal susceptibility testing methods published by the Clinical and Laboratory Standards Institute (M38, 3rd ed.) [29]. The ability to inhibit *Fusarium* biofilm formation using VOR, AmB, and 5-FC was assessed in a total of 8 isolates (Table 1) that represent variable susceptibilities to these three compounds. Antifungal solutions were prepared in RPMI-1640 medium as previously described at 2-fold serial dilutions ranging from 128 to 0.125 µg/mL. Then, 100 µL of these serial concentrations was distributed into each of 11 columns of a 96-well microplate, with the last column maintained without drugs to serve as a reference for maximum biofilm growth (control). Simultaneously, 100 µL of the stock spore suspension PMI-1640 medium was added to each well. The microplates were incubated for 24 h at 37 °C, followed by OD_595_ measurement via CV assay, as described above. All samples were prepared in triplicate, and the mean value was used for comparison.

The percentage inhibition was calculated using the following equation:Percentage inhibition (%) = 100 − [(OD_595_ T2 − OD_595_ T1)/OD_595_ control × 100](1)
where OD_595_ control is the absorbance of the control wells (biofilm development without drugs), OD_595_ T2 represents the absorbance of wells after 24 h of incubation with antifungal agents, and OD_595_ T1 is the absorbance of wells before incubation. The minimum inhibitory concentration (MIC) of biofilm is defined as the minimum antifungal concentration required to inhibit ≥50% biofilm formation (MIC_50_) or ≥90% biofilm formation (MIC_90_). IC50 curves were constructed in GraphPad Prism 7.00 (GraphPad Software Inc., San Diego, CA, USA) using non-linear regression inhibitor vs. dose response.

### 2.9. Biofilm Susceptibility of Combination Treatment

To evaluate the efficacy of a combination antifungal treatment (VOR, AmB, and 5-FC) in preventing biofilm growth, the 3 antifungal agent suspensions were combined to prepare 2-fold serial dilutions totaling 128–0.125 µg/mL. Following incubation for 24 h at 37 °C, biofilm growth was quantified using the biofilm quantification CV assay. Experiments were performed in six wells with three repetitions. The brightfield microscopy images and EPS composition analysis with each fluorochrome were performed as described above in Section 2.3 and Section 2.5.

### 2.10. Statistical Analysis

Growth diameter data from each collected day were averaged from 21 strains of the *F. solani* species complex and 6 strains of the *F. fujikuroi* species complex for comparison with the other 2 species complexes. The significance of the growth diameter, growth inhibition following drug treatment, and the biofilm composition assay were analyzed using one-way ANOVA with Geisser–Greenhouse correction using GraphPad Prism 10 (Boston, MA, USA). The biofilm formation under different sugar sources was analyzed using two-way ANOVA followed by Dunnett’s multiple comparison test. IC50 values were determined using nonlinear regression. Statistical significance was indicated as follows: “*” denotes *p* < 0.05, “**” indicates *p* < 0.01, and “***” denotes *p* < 0.001 versus *F. solani* A1, unless otherwise indicated.

## 3. Results

### 3.1. High Temperatures Inhibit Fusarium Growth

At 30 °C, the average diameter (cm) of triplicate samples for each isolate demonstrated that members of the *Fusarium solani* species complex (n = 21) grew faster than those from six isolates of *Fusarium fujikuroi* (*p* < 0.01), as well as isolates from the other two species complexes (Figure 1A). Due to the rarity of the *Fusarium oxysporum* species complex (FOSC) and *Fusarium incarnatum-equiseti* species complex (FIESC) isolates in our collection, we were only able to include one isolate from each. Notably, the *F. incarnatum-equiseti* isolate exhibited slower growth compared to all other tested strains. Nevertheless, at 42 °C, the growth differences between *F. solani* and the three other species were not evident, as elevated temperature suppressed growth rates across all isolates, including those within the *F. solani* species complex (Figure 1B). By day 9, the maximal colony diameter at 42 °C was 0.6 cm in all isolates, which was only 1/10th of the growth observed at 30 °C.

### 3.2. Fusarium Morphology and Anaerobic Growth

Morphologically, colony pigmentation of *Fusarium* species varied distinctly on PDA at 30 °C: *F. solani*, *F. incarnatum-equiseti*, and *F. oxysporum* exhibited pale brown, orangish-brown, or purple hues in contrast to the white-cotton to grayish appearance of *F. fujikuroi* (Figure 2A). Reverse colony colors were orangish-brown in all species. Microscopic examination revealed notable differences in structures of conidiogenous cells and spore production among these species. *F. incarnatum-equiseti* displayed more branched conidiogenous cells with sparse macroconidia than others, while the other three species produced abundant macroconidia from single, long monophialides, along with a large number of scattered microconidia. The multicellular macroconidia of all four species ranged from 25 to 40 μM in size, with septa measuring 3 to 7 μM in length. Despite the differences in conidial abundance, the hyphal formations were similar among the four species complexes. Additionally, no chlamydospores were observed in the *F. solani* isolates under both growth conditions [30].

Long-term anaerobic growth is critical for opportunistic pathogen survival during host infection. Studies have shown that the deeper layers of bacterial biofilms are anaerobic [31]. To better understand the biofilm biology of *Fusarium* species, we analyzed the growth characteristics of various *Fusarium* spp. under oxygen-limiting conditions (Figure 2B). Compared to aerobic growth, filamentous growth is more pronounced under anaerobic conditions for all four species, particularly *F. fujikuroi* and *F. incarnatum*, which display extensive fluffy edge appearances (Figure 2B). With more filamentation in agar culture, the growth rate of *F. fujikuroi* is less inhibited, showing a 34% reduction, whereas the other three species exhibit greater inhibition, ranging from 50% to 55% (Figure 2C).

### 3.3. Time Course of Biofilm Formation in 2% Glucose RPMI Medium

Biofilms allow bacteria and fungi to survive under unfavorable conditions in their habitat. Molecular and microscopic evidence suggests the existence of a succession of biofilm phenotypes, including adherence, initiation, maturation, and cell dispersion [32].

We first determined the time course of biofilm formation within 72 h to determine the critical time point of biofilm maturation using the crystal violet method. The 29 *Fusarium* isolates include FSSC (n = 21), FFSC (n = 6), FIESC (n = 1), and FOSC (n = 1), which were cultured with an OD_595_ = 0.25 inoculum in RPMI-1640 medium supplemented with 2% D-glucose. Following incubation at 37 °C, isolates formed dense biofilms that adhered to the polystyrene surface by 24 h post-inoculation, displaying OD_595_ of 1.572 ± 0.202 for FSSC, 1.860 ± 0.081 for FOSC, and OD_595_ > 2.0 for FFSC and FIESC. The differences observed among the four species complexes may be attributed to strain variation rather than species differences, as demonstrated by the variable responses of *F. solani* strains to the 2% glucose medium. Within the *F. solani* group, strain A17 exhibited the highest biofilm biomass (OD_595_ > 2.0, *p* < 0.001), while strain A11 showed the lowest biomass (OD_595_ = 1.274, *p* < 0.01) compared to the reference strain A1 (OD_595_ = 1.490).

Nevertheless, as shown in Figure 3A, biofilm biomass in most of the 29 isolates reached their maximum optical density (OD) (1.5~2.0) as early as 24 h post-inoculation. Since there were no significant differences in biofilm biomass among the four species during the 48 to 72 h period, this suggests that the critical period for biofilm maturation in *Fusarium* may occur between 24 and 48 h. Therefore, we focused on biofilm growth at 24 h in subsequent experiments.

### 3.4. Biofilm Growth Under Different Sugar Sources

The formation of *Fusarium* spp. biofilms under the influence of different sugar media was further investigated by culturing the fungi in RPMI-1640 media supplemented with 0.2% glucose, 2% trehalose, 2% xylose, and 2% glycerol and comparing the results with growth in 2% glucose (control). We used the biomass of *F. solani* A1 (Table 1) of each sugar source as a reference for statistical comparison with those produced by other strains under each sugar condition.

Generally, a reduced glucose concentration or supplementation with other sugars decreased the biomass in all strains at 24-h, as shown in Figure 3B. Within the FSSC group, the B9 isolate demonstrated a significantly more developed biofilm compared to other strains grown with either 2% dextrose (*p* < 0.001), 0.2% glucose (*p* < 0.001), 2% xylose (*p* < 0.001), or 2% trehalose (*p* < 0.001), which is similar to the growth pattern of *F. incarnatum-equiseti* B5 in different sugar sources. Among the four *Fusarium* species, the total biomass produced by *F. oxysporum* remained comparatively unaffected by sugar supplementation, in contrast to the highly impacted biofilm formation of *F. solani* and *F. fujikuroi* (*p* < 0.001), as shown in Figure 3B.

Regarding biomass differences by sugar types, we found that xylose and glycerol had greater inhibitory effects on biomass deposition across all isolates tested. The reductions in biomass for *F. fujikuroi*, *F. incarnatum-equiseti*, and *F. oxysporum* were 43%, 37%, and 15%, respectively, under 2% xylose medium, while *F. solani* A17 and B3 showed a reduced biomass of 56–58% compared with their baseline biomass formation under 2% glucose medium.

In glycerol medium, the biomass was reduced by 40% in *F. solani* A11 and 44% in *F. solani* A17 compared to growth in 2% glucose. The reduced rates in *F. fujikuroi* B4, *F. incarnatum-equiseti* B5, and *F. oxysporum* B8 were 44%, 37%, and 8%, respectively. The lower effect of glycerol on *F. oxysporum* B8 biofilm warrants further investigation to understand pathogenic implications. In our previous studies, fungal growth in glycerol was used to predict mitochondrial metabolism in *Candida albicans* [33,34]. The variable growth capacities among *Fusarium* species under glycerol may reflect differences in mitochondrial dependency for carbon metabolism.

### 3.5. Biofilm Composition by Fluorescence Microscopy

The composition of a biofilm can change significantly over time. Early stage biofilm formation, defined by the first 24 h following inoculation (Figure 3A), may be particularly important to maximize the efficacy of clinical treatment strategies. The biofilm composition in four *Fusarium* species complexes was first assessed using fluorescence microscopy at 12 h and 24 h time points. The results in Figure 4A present the merged and individual fluorescence images of one strain of each species: *F. solani* A17, *F. fujikuroi* B4, *F. incarnatum-equiseti* B5, and *F. oxysporum* B8. The image of CFW and NR represents the polysaccharide content and lipids on the surface of cells and within the extracellular biofilm matrix. Hoechst 33342 measures the total DNA content in both living and fixed cells, providing an estimate of biofilm cell numbers. Rhodamine 123 stains serve as an indicator of protein content.

At 12 h, *F. fujikuroi* formed a denser mycelial structure than the other three species. *F. fujikuroi* biofilms produced more abundant polysaccharides. In contrast, the dense biofilms of *F. solani* and the fragmented biofilm of *F. oxysporum* demonstrated a higher amount of NR staining for lipids structurally composed of mycelium and conidia. No biofilm was formed in *F. incarnatum-equiseti* at 12 h due to the lack of a mycelial structure (Figure 4A).

By 24 h, biofilms had matured in *F. solani*, *F. fujikuroi*, and *F. oxysporum*, as demonstrated by the thick layers of mycelium that were heavily coated with polysaccharides in the merged and CFW images in Figure 4A. However, the mycelial network had only started to form in F. incarnatum-equiseti, and stronger staining with Nile Red and Rhodamine 123 gave its merged image a weaker blue appearance. The average composition from multiple microscopic imaging areas are summarized for each individual species in Figure 4B. Compared to the higher integrated densities of *F. solani* A17 in CFW, NR, and Rhodamine 123 stainings, the other three species produced fewer polysaccharides (*p* < 0.05). *F. fujikuroi* and *F. oxysporum* produced fewer lipids and proteins than other tested isolates. As there was no significant difference in cell numbers among the four species (Hoechst 33342), lipids and proteins constitute a larger portion of the biofilm in the early stage, which may be more associated with conidial morphology. Meanwhile, polysaccharides, aligned with mycelial structures, constitute the major portion of the biomass in mature biofilms.

### 3.6. Rapid Increase in Cell Numbers and High Lipid Levels at Early Biofilm-Formation Stages

To determine the critical timepoint for biofilm formation and to understand the dynamic changes in biofilm composition over a 24 h period, we extended the analysis of 10 isolates using CFW, Nile Red, Rhodamine 123, and Hoechst 33342 at 6 h intervals, measured using spectrophotometry. The number of strains per species is shown in Figure 5. Consistent with the fluorescent microscopy results, the DNA content rapidly increased, reaching a plateau by 12 h across all four species (top right panel). As cell numbers increased, the lipid content (Nile Red) also reached near-plateau levels by 12 h in at least three species (bottom right panel). In contrast, polysaccharide and protein levels increased more linearly over the first 24 h, with polysaccharides reaching 1.0 to 1.5 × 10^5^ MFI (Mean Fluorescence Intensity) by 24 h, compared to the control (identical cultures without fluorescent staining).

By 24 h, the biofilm composition—lipid, DNA, and protein contents—was similar among the four species. However, *F. fujikuroi* produced more polysaccharides than *F. oxysporum* and *F. incarnatum-equiseti*, consistent with the more intense blue tone observed at 12 h in *F. fujikuroi* (top left panel in Figure 4A). Although *F. solani* produced an equal amount of polysaccharides as *F. fujikuroi* by 24 h, its polysaccharide level was significantly lower at the 12 h time point (*p* < 0.05). These findings suggest that the first 12 h could be critical for establishing the fungal architecture and providing sufficient lipid content required for biofilm formation, while the continued increase in polysaccharides and proteins plays a key role in biofilm maturation.

### 3.7. Inhibition of Biofilm Formation by Antifungal Agents

Extended-spectrum azoles, such as VOR, along with classical systemic antifungals AmB and 5-FC, remain the first-line therapies for *Fusarium* infections in clinical settings despite reports of reduced susceptibilities worldwide. Their anti-biofilm activities were evaluated using the sessile MIC_50_ values (sIC_50_), defined as a 50% reduction in the metabolic activity of the biofilm compared to that of the control growth using the XTT assay. Eight isolates, including five *F. solani* isolates, were chosen to represent strains with variable planktonic MIC_90_ values (Table 1). The sIC_50_ values for the individual or combinations of the three antifungals are displayed in Figure 6. The left panel shows the results for the five *F. solani* isolates, while the right panel compares four different *Fusarium* species, all assessed 24 h post-inoculation.

The antibiofilm activities of VOR and AmB were evaluated across different species at concentrations ranging from 0.125 to 128 μg/mL (Figure 6). For VOR, an sIC_50_ value of 0.68 μg/mL was found in the VOR-susceptible *F. solani* A1, which is significantly lower than the sIC_50_ values of the other four *F. solani* strains (*p* < 0.01 to 0.001), all of which exhibited MIC_90_ values above 32 μg/mL. Compared to the relatively VOR-resistant *F. solani* A17, *F. incarnatum-equiseti* and *F. fujikuroi* showed similarly reduced sessile-inhibitory effects. Biofilm biomass formation was least inhibited in *F. oxysporum* B8 (*p* < 0.05), which had the highest sIC_50_ value (Figure 6A). Regarding the sessile IC_50_ values of AmB (Figure 6B), the sIC_50_ values for *F. solani* A17 and B9 and *F. fujikuroi* B4 were low, corresponding to their MIC_90_ values of 2 μg/mL, 8 μg/mL, and 0.25 μg/mL, respectively. In contrast, *F. solani* B3, *F. incarnatum-equiseti* B5, and *F. oxysporum* B8 behavior diverged significantly from the AmB-susceptible *F. solani* A17 and *F. fujikuroi* B4, showing antibiofilm inhibition ranges between 20% and 70%. Although these results suggest that the antibiofilm activities of VOR and AmB vary among different strains, planktonically VOR- and AmB-resistant strains generally exhibit greater biofilm resistance to selected antifungal agents with respect to their biofilm formation.

The anti-biofilm effect of 5-FC is much lower than that of VOR and AmB based on sIC50 values (Figure 6C). A minimum of 50% or higher biofilm inhibition required at least 128 μg/mL of 5-FC for any given species. The sIC50 values among *F. solani* strains ranged from 0.42 μg/mL (A17) to 15.28 μg/mL (A11), while values for non-solani species were 11.3 μg/mL for *F. fujikuroi* B4, 7.59 μg/mL for *F. incarnatum-equiseti* B5, and 2.3 μg/mL for *F. oxysporum* B8. These results suggest that achieving significant growth inhibition in non-*F. solani* species may be more difficult with 5-FC.

Despite the poor inhibition efficacy of individual drugs, the combination of the three antifungals significantly reduced the sIC50 values for all isolates (Figure 6D). As shown in the left panel of Figure 6D, the sIC50 values ranged from 0.33 μg/mL in A11 to 3.88 μg/mL in B9 among the five *F. solani* isolates and were less than 2 μg/mL for *F. fujikuroi* B4, *F. incarnatum-equiseti* B5, and *F. oxysporum* B8. Indeed, the antibiofilm effects of this combination resulted in no significant differences between any two given isolates (the right panel of Figure 6D).

### 3.8. Alteration of Biofilm Composition Under Antifungal Agents by Fluorescence Microscopy

To better visualize the antibiofilm effect of antifungals, we investigated changes in the biofilm composition of four *Fusarium* species (A17, B4, B5, and B8) at 8 μg/mL of VOR, AmB, 5-FC, and their combination through fluorescence microscopy. Here, *F. solani* A17, which is resistant to VOR and susceptible to AmB, was compared with other *Fusarium* species that exhibit higher resistance to both VOR and AmB.

At 8 μg/mL of antifungal agents, inhibitory effects were more evident in *F. incarnatum-equiseti* B5 and *F. oxysporum* B8 as early as 12 h post-inoculation. Interestingly, VOR demonstrated a stronger inhibitory effect on total biomass compared to AmB or 5-FC. Microscopy revealed fewer visible mycelial structures in non-*F. solani* species at 24 h with VOR. Biofilm reduction proved particularly challenging for *F. solani* A17, as more mycelial structures were observed under VOR at 24 h (Figure 7A, right panel). The effectiveness of AmB was more consistent across all four species despite its high sIC_50_ values in *F. incarnatum-equiseti* B5 and *F. oxysporum* B8 (Figure 6B). Across all tested *Fusarium* samples, 5-FC is generally less effective as an anti-*Fusarium* agent (Figure 7A).

The combination of VOR, AmB, and 5-FC at a final concentration of 8 μg/mL effectively suppressed biofilm formation by 48 h, regardless of the variable drug susceptibilities of each species. While the mycelial structure was severely inhibited in *F. solani*, it was absent in *F. fujikuroi* B4 and barely seen in *F. incarnatum-equiseti* B5 and *F. oxysporum* B8 at 24 h (Figure 7A, right panel). These results underscore the efficacy of combination therapy in controlling biofilm formation. Conversely, monotherapy with a high dose of VOR can effectively suppress biofilm formation in non-solani *Fusarium* species, whereas the effect of AmB is less pronounced.

Under 8 μg/mL of VOR treatment, polysaccharides were suppressed by nearly 50% across all species, with minor effects on protein and lipid contents in non-solani species at 24 hours (Figure 7B). AmB treatment can selectively reduce polysaccharides by more than 50% in *F. solani* A17 and *F. incarnatum-equiseti* and protein in *F. fujikuroi*. Compared to the stronger inhibitory effects of VOA and AmB, the influence of 5-FC on biofilm composition was generally about half as effective in reducing polysaccharides, proteins, and lipids.

Evidently, the combination of all three compounds had a significant advantage in biofilm suppression compared to each compound alone, as demonstrated by reductions of more than 50–60% in polysaccharides and lipids and a 40% reduction in protein contents (Figure 7B). Among the four species, *F. fujikuroi* and *F. incarnatum-equiseti* responded well to the combination treatment, while *F. oxysporum* was the least responsive, particularly in terms of inhibiting protein and lipid content, which was noted under microscopy without antifungals (Figure 4A). Taken together, VOR is more effective than AmB in controlling biofilm formation, especially in reducing polysaccharide content in strains that display VOR resistance. Despite the more rapid growth and denser biofilm formation of *F. solani* and *F. fujikuroi* (Figure 4A), lipids and polysaccharides can be effectively suppressed by VOR alone or in combination treatment. In contrast, *F. oxysporum* showed the lowest response to antifungals in terms of biofilm composition (Figure 7B), which aligns with its flexibility in sugar alteration (Figure 3B). This resilience in biofilm formation despite environmental changes may justify its stronger adaptation abilities, which could be significant for its clinical relevance.

## 4. Discussion

An understanding of biofilm biology is crucial for developing clinical interventions that reduce biofilm proliferation. While biofilm-acquired drug resistance is traditionally attributed to physical barriers provided by extracellular matrices (ECM) that limit drug penetration [35], emerging evidence highlights additional changes within the biofilm that extend beyond its physical properties. The interplay of the increased activity of drug efflux pumps, reduced ergosterol levels (the target of azoles and polyenes), and stress responses often occur at different stages of biofilm formation [36,37,38]. In clinical settings, the incomplete eradication of dormant fungal cells nested in nails and keratin may lead to chronic infections [39,40].

Biofilms have been implicated in many fungal pathogeneses, including *Aspergillus*, *Candida*., and *Fusarium* spp. [35,41,42]. In clinical settings, these fungal biofilms not only prevent antifungals from effectively reaching the site of infection but also impact fungal–host interactions [43]. Similar to bacterial biofilms, fungal biofilms on indwelling medical devices such as catheters have become an increasing clinical concern [44,45]. For instance, biofilms are linked to the development of fungemia due to the release of viable apical biofilm cells [46]. Interestingly, cells detached from mature biofilms have exhibited greater cytotoxicity than their planktonic counterparts [47].

To better understand the dynamics and metabolic dependencies of biofilm development in clinically relevant *Fusarium* spp., we examined biofilm formation in *Fusarium* isolates from four *Fusarium* species complexes over time and under varying sugar sources. The time course of biofilm formation in these species aligns with previous observations in an *F. solani* keratitis isolate by Córdova-Alcántara et al. [48]. Our data revealed that the biomass after 48 h was consistent across four species, with notable differences observed within the first 24 h, particularly at the 12 h mark. Based on the SEM results obtained by Córdova-Alcántara et al., cells at this early stage underwent germination, elongation, and hyphal development. Given that lipids rapidly accumulate within the ECM during this period (Figure 4 and Figure 5), we conclude that the initial 12 h of high lipid deposition may serve as a critical time point for preventing biofilm maturation.

Despite strain-specific biofilm characteristics, several general patterns emerged across the four clinically relevant species: (1) High glucose levels in the medium accelerated biofilm formation. (2) *F. oxysporum* exhibited less dependency on glucose for biofilm production. The robust growth of *F. oxysporum* under glycerol indicates its metabolic flexibility and a possible mitochondrial advantage over other species. In contrast, *F. fujikuroi* and *F. solani* failed to form biofilms under glycerol conditions. (3) Voriconazole-resistant strains produced more robust biofilms. (4) Extracellular lipids (stained using Nile Red) were the major components of early stage biofilms (12 h post-inoculation), while cell wall polysaccharides and extracellular proteins dominated in late-stage biofilms. These findings underscore the importance of understanding biofilm biology for developing effective antifungal strategies.

The compositional changes in biofilms are significant for several reasons. In *C. albicans* biofilms, lipids present on the plasma membrane and within the ECM are crucial for maintaining biofilm architectural stability [49]. Dynamic changes in lipid profiles were found to not only impact cellular shape and physiology [50] but also fungal–host interaction. The presence of lipids in early *Fusarium* biofilms may suggest similar initial biofilm-formation mechanisms to those seen in *C. albicans*, while the later dominance of polysaccharides contributes to biofilm establishment and maturation, aiding in the shielding of fungi from host immune defenses. Notably, earlier and more intense CFW staining (6–12 h post-inoculation) was observed in *F. fujikuroi* compared to *F. incarnatum-equiseti* and *F. oxysporum*. This could be attributed to *F. fujikuroi*’s pronounced hyphal formation under both aerobic and anaerobic conditions.

Two anatomical locations cannot be overlooked when discussing *Fusarium* infections: the nails and keratin-rich tissues, both of which are superficial and commonly affected by *Fusarium*. *Fusarium* is the leading fungal pathogen in fungal keratitis, with an estimated 1,051,787 cases globally each year [51,52]. Additionally, *Fusarium* nail infections have been increasing in prevalence, becoming the most common non-dermatophyte mold (NDM) pathogen [53] that complicates onychomycosis therapy [54]. *Fusarium* keratitis (keratomycosis) often results from eye trauma caused by organic matter, contact lens use, or ocular surgeries [40,52]. Biofilm formation has been confirmed on contact lenses and animal keratin [40,55,56], as well as in ex vivo human fingernails [57,58]. Of particular importance, the dormant fungal cells within these biofilms may significantly increase the risk of invasive infections later, particularly in individuals with compromised immune systems.

In contrast to highly thermoresistant environmental fungi like *Aspergillus* spp., which often display robust growth at 45 °C and thrive in internal organs such as the lungs, *Fusarium* species prefer to form biofilms in the nails and eyes. Our data show that growth at 42 °C is significantly reduced to approximately one-tenth of the rate observed at 30 °C for all isolates, regardless of their variable growth rates within these *Fusarium* species complexes at 30 °C. When simulating a host environment, anaerobic growth is inhibited by approximately 34% in *F. fujikuroi* and 55% in *F. solani* compared to their aerobic growth. However, the increased mycelial development under anaerobic conditions could be advantageous for tissue invasion. Surface sites may offer a more favorable environment for *Fusarium*, where temperatures remain below 37 °C and harbor sufficient oxygen levels. In comparison, internal tissues, with nearly anaerobic conditions typically ranging from 3 to 9% oxygen levels [59], present a more challenging environment for *Fusarium* species.

Azole antifungals used in agriculture likely affect *Fusarium* resistance in clinical settings as well [23]. Azole drug resistance is particularly of interest in our ongoing studies. Despite varying planktonic MIC_50_ values, we selected 8 μg/mL of VOR, AmB, 5-FC, and a combination of these three compounds to study their anti-biofilm activities. Our data indicated that while VOR at this concentration may not inhibit the planktonic growth of VOR-resistant strains, the antifungal agent still significantly inhibits biofilm formation, primarily by decreasing polysaccharide content. Compared to the other compounds, 5-FC exhibited a mild anti-biofilm effect, with notable activity only against *F. incarnatum-equiseti*. AmB demonstrated less anti-biofilm activity compared to VOR, corroborated by the finding that both *F. fujikuroi* and *F. oxysporum* had less than 20% and 0% suppression of polysaccharides, respectively. However, the combination treatment of VOR, AmB, and 5-FC significantly reduced biofilm biomass across all species, with the most pronounced effects observed in non-solani *Fusarium* species.

## 5. Conclusions

This study investigated the in vitro biofilm characteristics of four *Fusarium* species complexes in terms of their composition under antifungal treatments and associated biological traits. In conclusion, our study highlights the complexity and variability of biofilm formation among different *Fusarium* species. The data underscore the significance of thermotolerance, anaerobic growth capabilities, and biofilm composition in fungal pathogenicity. We demonstrated that biofilm composition evolves over time, with early stage biofilms primarily composed of lipids and late-stage biofilms dominated by polysaccharides and proteins. Our findings also revealed that antifungal treatments, particularly combinations, significantly altered the biofilm structure and composition. This underscores the potential for developing more effective treatment strategies, pending confirmation in in vivo biofilm models. Understanding these factors is crucial for creating targeted interventions against *Fusarium* infections, especially in clinical settings, where biofilm characteristics can significantly impact treatment outcomes.

## 6. Limitations

Species bias is a significant limitation in this study. *F. solani* was the most common in our collection, followed by *F. fujikuroi* and *F. oxysporum*. Despite numerous attempts, we faced challenges with incomplete collection data associated with the CDC serial strains. The more potent biofilm formation and pronounced hyphal development of *F. fujikuroi* observed in this study might explain its higher prevalence as a secondary pathogen.

Even with one *F. oxysporum* isolate, our data show unique pathogenic advantages of this species, including metabolic flexibility, which reduces its dependence on sugar availability, which may allow it to thrive under varying nutrient conditions within the host. However, due to sample limitations, we cannot definitively conclude whether the discrepancy is due to geographical sampling differences. Further research with a larger number of clinical samples is needed to explore geographical and pathogenic variations among *Fusarium* species.

Additionally, we acknowledge the limitations of in vitro models not fully replicating the complexity of clinical biofilms. Further validation using in vivo models will be essential to better assess the clinical relevance of these findings.

## Figures and Tables

**Figure 1 jof-10-00766-f001:**
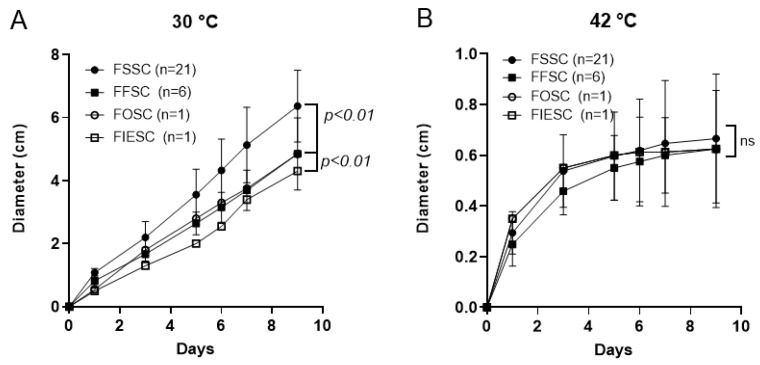
*Fusarium* colony growth at different temperatures. (**A**) Colony diameters of a total of 29 clinical isolates in *F. solani* species complex (FSSC), *F. fujikuroi* species complex (FFSC), *F. oxysporum* species complex (FOSC), and *F. incarnatum-equiseti* species complex (FIESC) on PDA at 30 °C. The number of isolates used in each complex is indicated by “(n = XX)”. Colony growth in the FSSC group is faster than in the FFSC, FOSC, and FIESC groups, *p* < 0.01. (**B**) Growth on PDA at 42 °C. The same set of clinical isolates showed a maximal diameter of 0.6 cm by day 9, in contrast to 6 cm at 30 °C. No significant (ns) difference was observed among the colonies of 29 isolates at 42 °C, with *p* > 0.05.

**Figure 2 jof-10-00766-f002:**
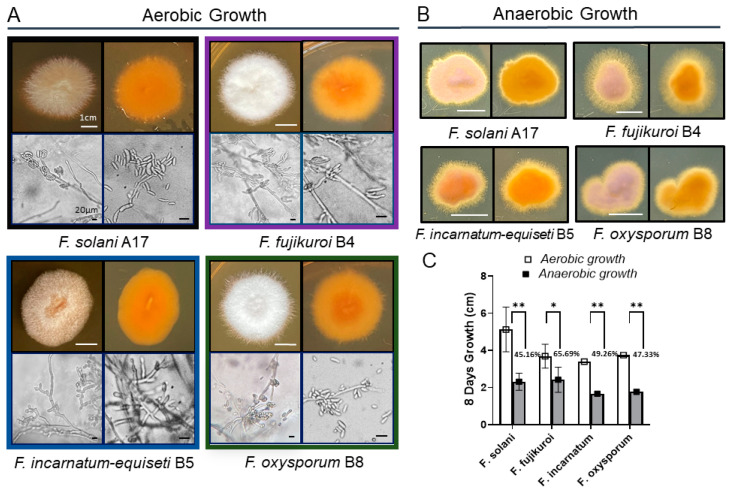
(**A**) Aerobic growth on PDA at 30 °C after 8 days, as demonstrated by *F. solani* A17, *F. fujikuroi* B4, *F. incarnatum-equiseti* B5, and *F. oxysporum* B8. For each species panel, the top row displays the colony morphology from the front (**left**) and reverse view (**right**). The bottom row displays the microscopic features of each species at low (**left**) and high magnification using brightfield microscopy, highlighting the differences in the shape and quantity of characteristic macroconidia among the four species. The white scale bar represents 1 cm, and the black scale bar represents 20 μM. (**B**) Morphological appearance of peripheral mycelial colonies, especially in *F. fujikuroi* and *F. incarnatum-equiseti* isolates, under anaerobic growth on PDA at 30 °C. (**C**) Growth inhibition of each of the 29 clinical isolates under anaerobic conditions. When compared to aerobic growth, an average growth inhibition of 34% was observed in *F. fujikuroi*, and 50% to 55% inhibition was observed in the other three species. Significant differences between aerobic and anaerobic conditions within each species are indicated by “*” for *p* < 0.05 and “**” for *p* < 0.01.

**Figure 3 jof-10-00766-f003:**
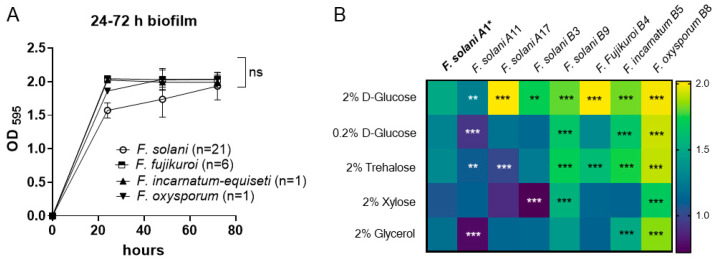
Biofilm formation following incubation with different sugar sources: (**A**) Time course of biofilm formation of the 29 *Fusarium* isolates in polystyrene plates in 2% glucose RPMI-1640 was assessed using the crystal violet method. No significant differences were observed among different species between 48 and 72 h. (**B**) Heatmaps of biofilm formation conducted by the 8 *Fusarium* strains grown in different carbon sources for 24 h post-incubation. The carbon sources and their concentrations in RPMI-1640 are indicated on the left side of the heatmap. The color bar on the right represents the OD_595_ optical density of each species of biofilm. Reduced levels of biofilm formation in 0.2% glucose or other non-glucose carbon biofilms are represented by blue to dark blue colors, in contrast to the biofilm formed in 2% glucose RPMI-1640 medium (**top row**). Stars denote significant differences between each strain and *F. solani* A1 (bold) in each carbon source condition, with “**” indicating *p* < 0.01 and “***” indicating *p* < 0.001, and “ns” as *p* > 0.05.

**Figure 4 jof-10-00766-f004:**
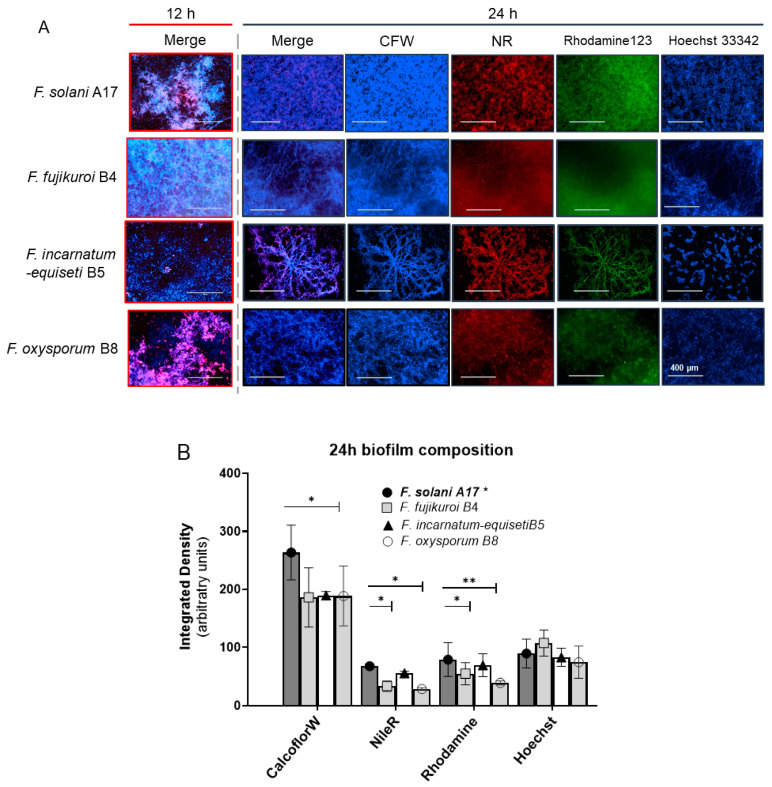
Early biofilm composition in different *Fusarium* spp. analyzed using fluorescence microscopy. (**A**) The composition of biofilm at 12 and 24 h post-inoculation. The isolates used in each species complex are presented in the left column. The abundance of polysaccharides and lipids in the extracellular matrix was assessed using Calcofluor White (CFW) and Nile Red (NR). Rhodamine 123 and Hoechst 33342 were used to evaluate protein and DNA contents within the biofilm. The merged images are a combination of CFW, NR, and Rhodamine staining. The scale bar is 400 μM. (**B**) The integrated density of each composition at the 24 h time point was across five microscopic fields (10× objective magnification) for each isolate. The relative levels of each composition in *F. fujikuroi*, *F. oxysporum*, and *F. incarnatum-equiseti* were individually referred to those shown in *F. solani* A17. Stars denote significant differences between A17 (bold) and other species, with “*” indicating *p* < 0.05 and “**” indicating *p* < 0.01.

**Figure 5 jof-10-00766-f005:**
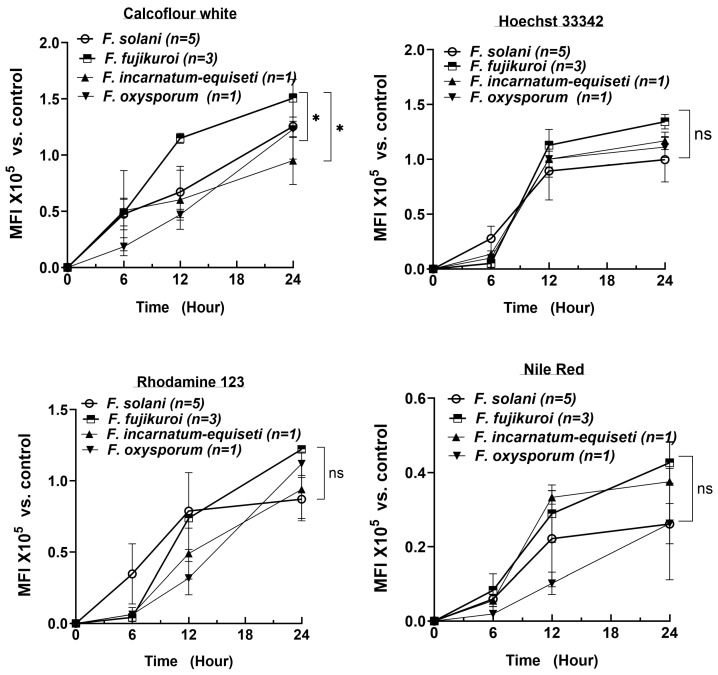
Biofilm composition at early stages (12 h post-inoculation) in four *Fusarium* complexes (n = 10 isolates). Fluorescent signaling, indicated by mean fluorescent intensity (×10^5^) on the *y*-axis, was measured at 6 h intervals using a fluorescent plate reader (Dynex Technologies). Four fluorochromes were used to measure different biofilm components: CFW for polysaccharides, Hoechst 33342 for DNA, Rhodamine 123 for proteins and indicators of cell metabolic activity, and NR for lipids secreted at the biofilm surface and within living cells. The control for each isolate was the parallel growth at each time point without the respective fluorescent dye. Significant differences in MFI between two species at 24 h is indicated by “*” for *p* < 0.05, “ns” for *p* > 0.05.

**Figure 6 jof-10-00766-f006:**
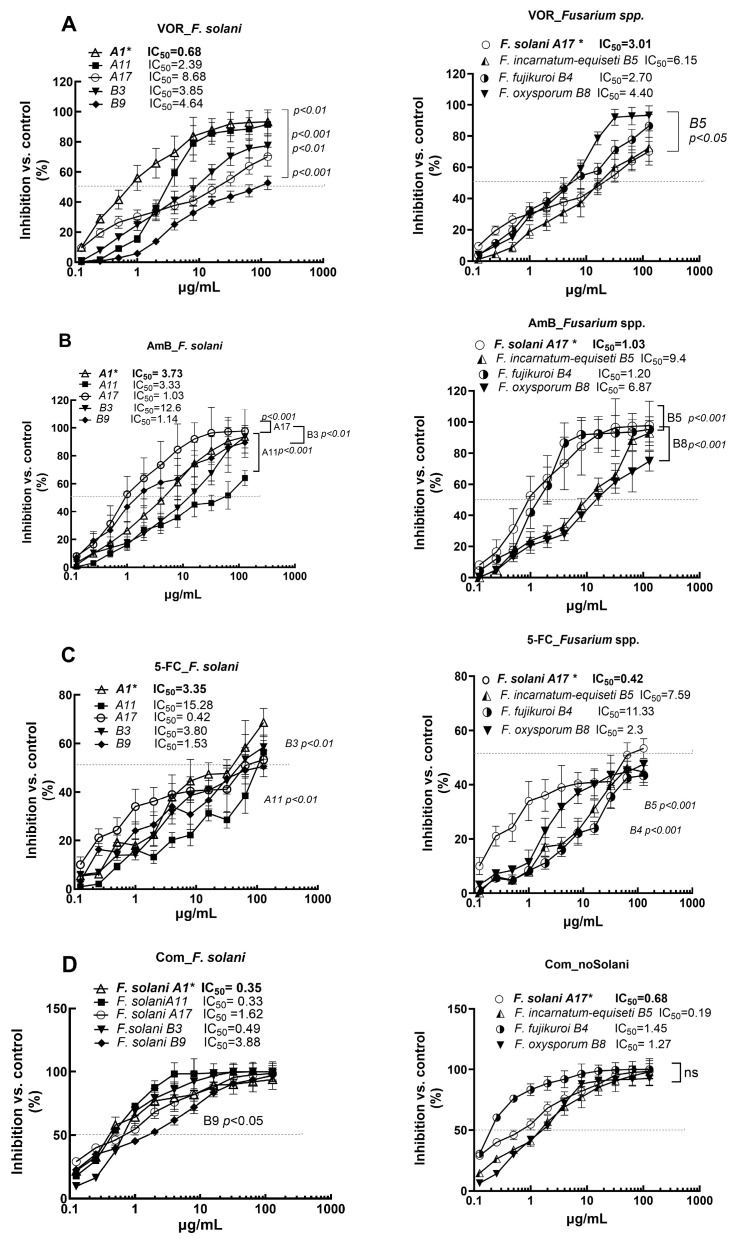
Anti-biofilm activities of voriconazole (panel **A**), voriconazole B (panel **B**), 5-fluorocytoscene C (panel **C**), and the combination of three compounds (panel **D**) at concentrations ranging from 0.13 to 128 μg/mL. The biofilm-inhibitory effect of each compound was compared with the XTT metabolic activity of the control well (without antifungal compounds). The left panel relate to the five *F. solani* isolates, with significance indicators showing the differences between A1 (bold) and the other *F. solani* isolates. The right panel shows significant differences between *F. solani* A17 and other *Fusarium* strains. The IC_50_ represents the concentration required for a 50% sessile-inhibitory effect of each compound on a given isolate. The *F. solani* A17*, in bold, served as the index strain for comparison and “ns” for *p* > 0.05.

**Figure 7 jof-10-00766-f007:**
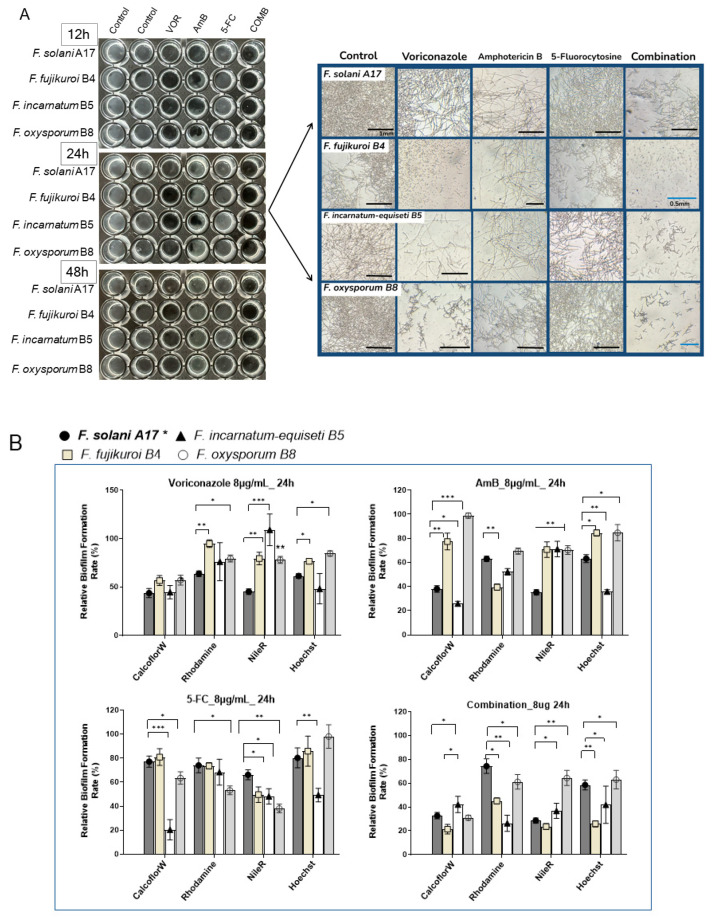
(**A**) The total biomass and biofilm structure of *Fusarium* spp. in the presence of an intermediate dose (8 μg/mL) of mono- and combination treatments of voriconazole, amphotericin B, and 5-fluorocytosine were evaluated. **Left panel**: biomass increases over 12–48 h with each compound at 8 μg/mL, with two control columns representing growth without antifungal agents. **Right panel**: morphological views of mycelial structure and conidiospores 24 h after initial treatment under antifungal conditions. The black scale bar represents 1 mm, and the blue scale bar represents 0.5 mm. (**B**) The compositional changes under each antifungal or combination treatment at 24 h versus the control without drug were assessed using their fluorescent integrated densities. The *F. solani* A17*, in bold, served as the index strain for comparison. “*” indicates *p* < 0.05, “**” indicates *p* < 0.01, and “***” indicates *p* < 0.001.

**Table 1 jof-10-00766-t001:** Eight clinically relevant *Fusarium* isolates and the planktonic MIC_90_ values for voriconazole (VOR) and amphotericin B (AmB).

Strain	Source	Species (Species Complex)	VOR MIC_90_ (µg/mL)	AmB MIC_90_ (µg/mL)
A1	CDC BO5546	*F. solani* (FSSC)	1	16
A11	CDC BO9398	*F. solani* (FSSC)	32	16
A17	CDC BO9407	*F. solani* (FSSC)	128	2
B3	Texas DI23-9	*F. falciforme* (FSSC)	128	2
B9	Texas DI23-15	*F. petroliphilum* (FSSC)	128	8
B4	Texas DI23-10	*F. fujikuroi* (FFSC)	128	0.25
B5	Texas DI23-11	*F. incarnatum-equiseti* (FIESC)	128	16
B8	Texas DI23-14	*F. oxysporum* (FOSC)	128	16

## Data Availability

The raw data supporting the conclusions of this article will be made available by the authors upon request.

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
