# Peer review of "Biofilm Formation in Clinical Isolates of Fusarium"

_jof, 2024, doi:10.3390/jof10110766_

Round 1
Reviewer 1 Report
1. Differences Between Experimental Complex Composition and Real Clinical Co-Infections
2. The Necessity of More Complex Infection Models to Validate Drug Combination Effects
3. Lack of Exploration into Post-Biofilm Disruption Treatment Strategies
4. Limited Clinical Relevance of Carbon Source Research
1. As mentioned in the discussion, the composition of the complex in the experiment and the actual clinical complex infection (filamentous fungi, bacteria, yeast) may be quite different, so that the experimental results are not more closely integrated with the actual clinical infection.
2. Combine with more complex infection models: biofilms may show different resistance in the human body at the site of infection (e.g. eye or skin) or on implants. Attempts can be made to validate the effects of drug combinations in materials closer to clinic (e.g. fungal contamination of medical devices) or in animal models to further increase the extrapolation and utility of experimental results.
3. Although the experiments concluded that biofilm weakening was achieved, there was a lack of exploration of subsequent therapeutic strategies. For example, after biofilm disruption, is it necessary to increase the dose of antibiotics or combine other therapies?
4. The effect of different carbon sources on biofilm formation was studied in the experiment. But the main aim of the experiment was to study the inhibitory effect of the drug on the biofilm. Please provide additional information on the relevance of carbon sources to the study aims.
Author Response
- As mentioned in the discussion, the composition of the complex in the experiment and the actual clinical complex infection (filamentous fungi, bacteria, yeast) may be quite different, so that the experimental results are not more closely integrated with the actual clinical infection.
Response: You raise an important point regarding the complexity of biofilms in clinical infections, where multiple organisms, including filamentous fungi, bacteria, and yeast, may contribute to the infection. We share this concern. It is well known that antibiotic treatment can increase the risk of invasive fungal infections, and in cases of nail onychomycosis, treatment failures are often attributed to mixed infections involving dermatophytes and Fusarium species. While dermatophytes typically respond to antifungal treatments, the inherently resistant Fusarium can emerge as a dominant pathogen due to its robust biofilm formation, as demonstrated in this study.
We acknowledge the limitations of in vitro models in fully replicating the complexity of clinical biofilms. Currently, developing techniques to accurately reflect biofilm composition in patients remains challenging. Until such methods become available, in vitro conditions—such as incubation at host temperature (37°C), low glucose levels, and relatively anaerobic environments—serve as a useful, albeit imperfect, approach to understanding the biological characteristics of Fusarium biofilms.
Unlike Candida albicans, which is often associated with invasive infections of endogenous origin, Fusarium infections tend to occur more frequently on external surfaces, such as the eyes and nails, with invasive infections being less common. This distinction further motivated our focus on understanding the biofilm biology of Fusarium, especially in the context of its role in surface infections.
Following your suggestion, we have revised the final part of the Conclusion and Limitations section (in red) to address these concerns.
- Combine with more complex infection models: biofilms may show different resistance in the human body at the site of infection (e.g. eye or skin) or on implants. Attempts can be made to validate the effects of drug combinations in materials closer to clinic (e.g. fungal contamination of medical devices) or in animal models to further increase the extrapolation and utility of experimental results.
Response: Thank you for this insightful suggestion. Our current study is an initial step in understanding the correlation between biofilm formation and resistance in Fusarium. We agree that biofilms may exhibit different resistance properties depending on the infection site, such as the eye, skin, internal organs, or implants. This has motivated us to expand our experiments by testing biofilm formation under varying conditions, such as different carbon sources and anaerobic environments. We plan to incorporate more complex infection models in future studies, such as fungal contamination of medical devices or animal models, to validate the effects of antifungal combinations on biofilms. This approach will help us better reflect clinical conditions, allowing for a deeper understanding of in vivo biofilm behavior and improving the clinical utility of our experimental results.
- Although the experiments concluded that biofilm weakening was achieved, there was a lack of exploration of subsequent therapeutic strategies. For example, after biofilm disruption, is it necessary to increase the dose of antibiotics or combine other therapies?
Response: Thank you for your thought-provoking questions and insightful comments regarding antifungal influence on biofilm development and subsequent therapeutic strategies. Although our use of a fixed concentration of 8 µg/mL for each antifungal drug offers valuable insights into biofilm susceptibility (Figure 7), determining the optimal dosage—particularly considering factors such as drug toxicity in vivo (e.g., amphotericin B at this concentration)—requires further exploration in more clinically relevant models.
We acknowledge that the in vitro nature of this study limits the extrapolation of our findings to clinical settings, particularly in terms of therapeutic efficacy. The potential necessity for higher antibiotic doses or combination therapies post-biofilm disruption is yet to be confirmed in in vivo biofilm models. These models, which better mimic the complexity of biofilm formation and drug interactions within living systems, would provide a more appropriate platform to address your concerns regarding therapeutic strategies after biofilm weakening. This represents a crucial direction for our future research.
- The effect of different carbon sources on biofilm formation was studied in the experiment. But the main aim of the experiment was to study the inhibitory effect of the drug on the biofilm. Please provide additional information on the relevance of carbon sources to the study aims.
Response: Thank you for your insightful question. The investigation of different carbon sources on biofilm formation (Figure 3B) was intended to differentiate the carbon utilization capacity between drug-resistant and susceptible strains during biofilm formation. The results indicated that azole-resistant Fusarium solani and three other species exhibit greater nutrient flexibility compared to azole-susceptible F. solani strains. Notably, F. oxysporum showed less constrained by carbon source availability.
While the primary objective of this study was to evaluate the inhibitory effects of antifungal drugs on biofilm formation, exploring carbon source utilization still offers important context. It highlights Fusarium's adaptability and potential for biofilm formation in diverse clinical environments. Therefore, this aspect can be considered a secondary aim of the study: investigating the biology of Fusarium growth and biofilm formation.
Clinically, invasive Fusarium infections are less common than those caused by environmental molds like Aspergillus. This may be due to Fusarium's reduced growth under anaerobic conditions, high temperatures, and limited nutrient availability in certain anatomical sites. Our observations with Candida albicans suggest that the ability to metabolize non-glucose carbon sources plays a crucial role in pathogenesis, particularly in tissues where glucose levels are low. Understanding Fusarium's adaptability to various environments, including different oxygen levels and carbon sources, complements the study’s broader goal of assessing biofilm inhibition by antifungal treatments.
Reviewer 2 Report
In this article, the authors investigate the development of biofilms of Fusarium spp. in relation to different carbon sources, incubation conditions and exposure to antifungal substances. The article is excellently written, with a very nice introduction, clear results and an informative discussion.
A few small remarks:
Lines 216 - 217: The concentrations of the stock solutions should be given either in mg/L or µg/mL to make the text clearer.
Lines 264 - 269: The text belongs in the Methods and not in the Results. Please correct it.
Lines 280 - 282: The text belongs in the Discussion and not in the Results. Please correct it.
Lines 318 - 319: The text belongs in the Discussion and not in the Results. Please correct it.
Lines 328 - 333: Move the text either to the Introduction or to the Discussion.
Figure 6: Replace 5-FC with 5-fluorocytosine.
Lines 571 - 577: The text belongs in the Discussion and not in the Results. Please correct.
Author Response
Major comments
Comments 1: In this article, the authors investigate the development of biofilms of Fusarium spp. in relation to different carbon sources, incubation conditions and exposure to antifungal substances. The article is excellently written, with a very nice introduction, clear results and an informative discussion.
Response: We really appreciate the reviewer for valuable comments.
Detail comments
A few small remarks:
Lines 216 - 217: The concentrations of the stock solutions should be given either in mg/L or µg/mL to make the text clearer.
Response: Thank you for the suggestion. We have updated the text to specify the stock concentration as 5000 µg/mL on page 5, line 217 for clarity.
Lines 264 - 269: The text belongs in the Methods and not in the Results. Please correct it.
Response: Thanks for pointing it out. We have moved the entire paragraph to the Methods section and combined it with Section 2.2.
Lines 280 - 282: The text belongs in the Discussion and not in the Results. Please correct it.
Response: Thanks for suggestion. The text from lines 280-282 has been moved to the Discussion section and is now located in lines 632-636.
Lines 318 - 319: The text belongs in the Discussion and not in the Results. Please correct it.
Response: We appreciated your suggestion; however, we prefer to retain this sentence in lines 311-312. This sentence serves to initiate the concept on studying the significance of anaerobic growth.
Lines 328 - 333: Move the text either to the Introduction or to the Discussion.
Response: Thank you for the suggestion. This sentence is intentionally included to introduce the focus of this part of the experiment. To maintain the flow of the manuscript, we would like to retain the first part of this paragraph in lines 321-323.
Figure 6: Replace 5-FC with 5-fluorocytosine.
Response: Thanks for pointing it out. We have made the change in the caption of Figure 6 to replace 5-FC with 5-fluorocytosine.
Lines 571 - 577: The text belongs in the Discussion and not in the Results. Please correct.
Response: Thank you for pointing this out. The content in lines 571-577 serves as a summary of the results in Section 3.7 regarding the inhibition of biofilm formation by antifungal agents. We prefer to retain it in lines 559-567 for continuity.